# Beyond Tokens: Rich Character Embedding for Low-Resource and Morphologically Rich Languages

**Felix Schneider** *felix.schneider@uni-jena.de*
*Computer Vision Group*
*Friedrich Schiller University Jena*

**Maria Gogolev** *maria.gogolev@uni-jena.de*
*Computer Vision Group*
*Friedrich Schiller University Jena*

**Sven Sickert** *sven.sickert@uni-jena.de*
*Computer Vision Group*
*Friedrich Schiller University Jena*

**Joachim Denzler** *joachim.denzler@uni-jena.de*
*Computer Vision Group*
*Friedrich Schiller University Jena*

**Reviewed on OpenReview:** *https://openreview.net/forum?id=4n4db5qmXZ*

## Abstract

Tokenization and sub-tokenization based models like word2vec, BERT and the GPTs are the state-of-the-art in natural language processing. Typically, these approaches have limitations with respect to their input representation. They fail to fully capture orthographic similarities and morphological variations, especially in highly inflected and under-resource languages. To mitigate this problem, we propose to computes word vectors directly from character strings, integrating both semantic and syntactic information. We denote this transformer-based approach Rich Character Embeddings (RCE). Furthermore, we propose a hybrid model that combines transformer and recurrent mechanisms. Both vector representations can be used as a drop-in replacement for dictionary- and subtoken-based word embeddings in existing model architectures. It has the potential to improve performance for both large context-based language models like BERT and small models like word2vec for under-resourced and morphologically rich languages. We evaluate our approach on various tasks like SWAG, declension prediction for inflected languages, and metaphor and chiasmus detection for various languages. Our experiments show that it outperforms traditional token-based approaches on limited data using OddOneOut and TopK metrics.

## 1 Introduction

As humans, we see the words *quality* and *qualification* as connected - even if we do not know the meaning of *qualification*, we can infer its meaning by its similarity to the word *quality* and due to the context it appears in. However, in a purely dictionary-based input this conclusion cannot be drawn; only context is available, the spelling similarities are not included. Methods like WordPiece tokenization can mitigate this to some extent. However, the way tokenizers split words into subtokens is not guaranteed to reflect the linguistic structure of the words, and can be suboptimal especially in multilingual modes (Rust et al., 2021) or when not trained on a sufficiently large corpus. For example, a standard English WordPiece model for BERT may tokenize *quality* as a single token, [quality]. At the same time, *qualification* can be tokenized into [qual], [##ifi], and [##cation] This again leads to the same problem of not having a concept of the similarity between *qual* and *quality* in the input. Figure 1 shows this in more detail.

Yet another problem are different spellings. For example, the following deliberately mistyped sentence part highlights that *tihs setnence can eeasily be raed by a hunam.* (Rayner et al., 2006). By contrast, a dictionary-based model needs explicit exposure to such variants in the training data (Clark et al., 2022; Tay et al., 2022; Xue et al., 2022). Without this, sensible vector representations for those words cannot be learned. Finally, likely owing to the English language focus of much of NLP research, dictionary-base approaches work well for languages like English, which are neither agglutinative nor highly inflected. Character- or byte-level approaches for contextual models exist, but they substantially increase input sequence length, reducing the effective context available to the model (Clark et al., 2022; Tay et al., 2022; Xue et al., 2022). Older LSTM-based forget context over long sequences (Pascanu et al., 2013), while modern transformer-based have memory requirements that grow quadratically with input legth (Vaswani et al., 2017). This limits the feasible sequence length. Since massive corpora improve robustness and coverage (Brown et al., 2020), for dictionary-based models the common mitigation strategy for spelling variations and rare words is being trained on extremely large corpora, hoping that sufficient spelling variants are present in context to learn meaningful representation.

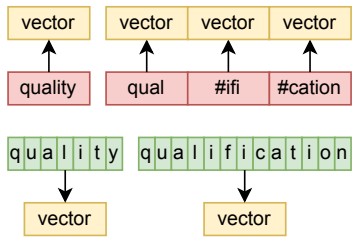

Figure 1: This figure shows an example for the limitations of subtokenization based embedding and our solution. The words are shown first under subtokenization and then under our approach. The tokens into which *qualification* is split is split are distinct to the token for *quality*. Also, one word is split into three vectors. In contrast, our novel approach computes a single vector representation for each word and accounts for spelling similarity.

These issues are magnified for low-resource and under-resourced languages. While large corpora exist for widely used languages (e.g. Mandarin, English, Spanish, German), languages with few speakers (e.g. Faroese) or historical languages (e.g. Middle high German), often only have small corpora (Joshi et al., 2020). Because many of those languages, such as those in the Indo-European family, are highly inflected, learning a separate representation for every inflected form is infeasible for single word tokens, and is prone to fail with subword-based tokenization (Rust et al., 2021). Related languages nevertheless share vocabulary and spelling similarities, even when the exact spellings differ (Snæbjarnarson et al., 2023). Figure 2 shows an excerpt of a text in Middle High German with translations into German and English (all Germanic languages). The similarities, especially between German and Middle High German, are notable; even in English some words remain recognizable despite Romance influence. In a typical token- or subtoken-based approach, however, all these words would be encoded as distinct tokens, with only limited or inconsistent input-level signal of their similarity.

*mgh*
Daz liet, daz wir hie wirken, daz sult ir rehte merken.

*de*
Das Lied, das wir hier wirken, das sollt ihr recht merken.

*en*
The song that we here create, that should you rightly remember.

Figure 2: Excerpt of a text in Middle High German (mgh), a low-resource language. Below it, the German (de) and English (en) translations.

Given these shortcomings, we propose a different approach. Instead of a dictionary of tokens or subtokens, our *Rich Character Embedding* uses a transformer-based neural network to compute a vector representation for a word directly from its character string. This vector can be used used as a drop-in replacement for dictionary- and subtoken-based word embeddings in existing model architectures.

In this paper, we present a novel transformer-based word vector representation computed from character strings that fulfills several criteria: (i) words that frequently appear in similar contexts have similar vector representations, comparable to traditional dictionary-based approaches; (ii) words with similar spellings, such as inflected variants or alternative spelling, yield similar vectors; (iii) the representation retains information about the original character sequence, facilitating inference of grammatical features such as case and number in inflected languages; and (iv) rare or unseen words can be encoded without subtokenization by leveraging learned similarities to related spellings. Finally, the model is designed to be easily integrated as a plug-

in input replacement for larger context-based language models such as BERT without extending input sequence length as character- or byte-level encoders do. We make our training code and datasets availabble at `https://www.github.com/cvjena/RichCharacterEmbedding`.

## 2 Related Work

For languages with large training corpora, vector-based text representations are commonly created using large transformer models such as BERT (Devlin et al., 2019), while GPT-like models are used for tasks such as text generation (Radford et al., 2018; 2019; Brown et al., 2020). However, training such large models is challenging for under-resourced languages due to limited corpora (Wu & Dredze, 2020). Multilingual variants like mBERT aim to address this but often favor high-resource languages, yielding suboptimal tokenization and representations for low-resource languages (Wu & Dredze, 2020).

Traditional word embeddings such as word2vec (Mikolov et al., 2013) and FastText (Bojanowski et al., 2017) provide an alternative that remains relevant for under-resourced languages due to their simplicity and lower data requirements (Grave et al., 2018). FastText in particular improves over word2vec by incorporating subword information, which helps to mitigate issues with out-of-vocabulary words and morphologically rich languages. However, both methods still require substantial corpora to learn meaningful representations, and their lack contextualization limits their effectiveness (Coto-Solano, 2022).

Another line of work addresses rare and out-of-vocabulary words by using lexical resources rather than only surface form or corpus context. Ruzzetti et al. (Ruzzetti et al., 2022) derive embeddings for rare words from traditional dictionary definitions, proposing DefiNNet and DefBERT as definition-based methods for OOV representation. This approach is complementary to ours: dictionary-definition methods use external semantic descriptions when available, whereas RCE derives representations directly from the character string and training corpus. Thus, RCE targets settings where orthographic and morphological regularities are informative, while definition-based methods rely on the availability and quality of lexical definitions.

For static single-word embeddings, the field has explored convolution-based enhancements that incorporate subword structure and convolutional layers to improve representations (Cao & Lu, 2017). These methods aim to capture syntactic and morphological information more effectively while maintaining efficiency. Nevertheless, evaluating of word embeddings in low-resource settings remains challenging, as many standard benchmarks were designed for high-resource languages and may not reflect the linguistic characteristics of under-resourced languages (Coto-Solano, 2022). The Odd One Out and TopK metrics address this issue by providing a more nuanced evaluation of word embeddings better suited to low-resource languages (Stringham & Izbicki, 2020).

Various approaches also make the aforementioned architectures more feasible for low-resource languages, including fine-tuning BERT with active learning (Grießhaber et al., 2020). In contrast, our approach directly targets the input representation of words for both word2vec-like and BERT-like models, serving as a drop-in replacement for traditional dictionary-based word embeddings. Because the rest of the model architecture remains unchanged, our method can be combined with these techniques to further enhance performance on low-resource languages.

## 3 Method Overview

Our novel approach *Rich Character Embedding* (RCE) computes a vector representation for words from their character strings. We represent every character as a one-hot-vector and use modifiers to represent features such as uppercase and diacritics (e.g. umlauts).

### 3.1 Basic Definitions

We define a training corpus $T$ as a collection of tokens $t_1, t_2, \ldots, t_k$ with $k$ tokens in total in the corpus. We tokenize by whitespace, so tokens represent the different words in the corpus; we use subtokens. Each token has a character string $C(t_n) = (c_{n1}, c_{n2}, \ldots, c_{nl})$ of length $l$. All characters $c$ belong to an alphabet

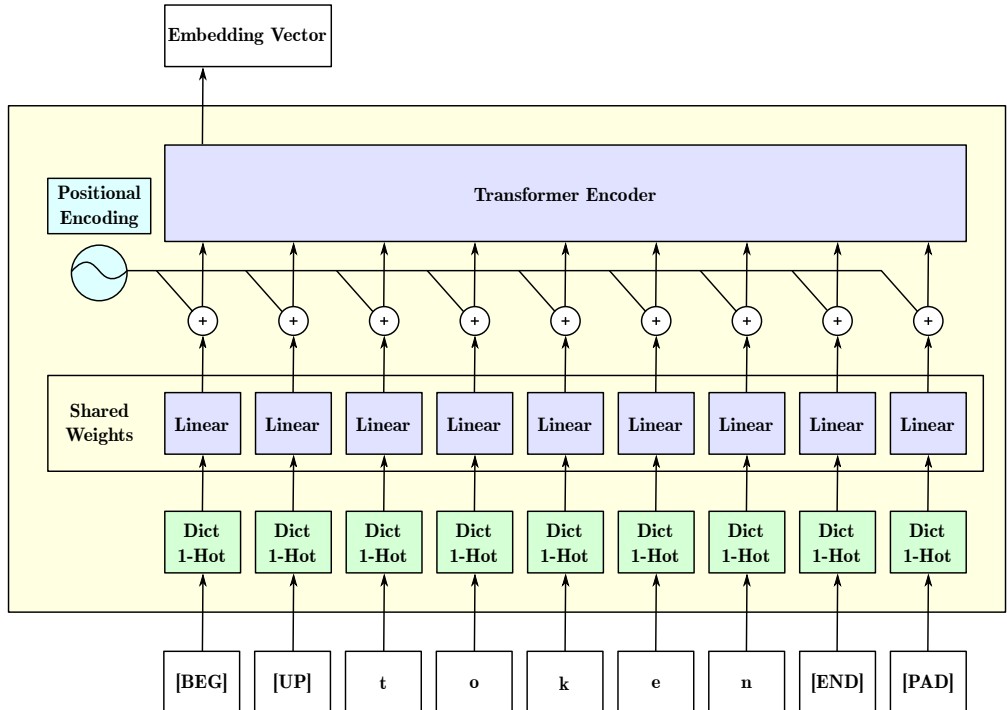

Figure 3: This figure shows the model architecture for the Rich Character Embedding. The input token (e.g., *Token*) is represented as its character string, with the capital $T$ split up into an [UP] modifier and the base character $t$. A transformer encoder maps the sequence to a vector representation. The first output (corresponding to [BEG]) is used as the word embedding.

dictionary $A$, which also contains additional entries as explained in section 3.2. The embedding vector of $C(t_n)$ is denoted $e(t_n)$.

If we train with one-hot-token prediction (Section 3.4), we build a dictionary $D_T$ for the most common tokens in the corpus $T$. Thus, a subset of the tokens $t_n$ in the corpus $T$ have a dictionary entry $D_T(t_n)$ and $D_T$ is a subset of the set of all unique tokens in the training corpus.

We denote $E_T$ as the embedding model trained on $T$. The input to $E_T$ is the character string $C(t_n)$ instead of the dictionary entry of $D_T(t_n)$. Thus, we can train on variants such as alternative spellings, inflection, suffixes, and prefixes in the training data without overhead or additional dictionary entries. If we train with one-hot-token prediction instead of or additional to character string prediction, we need at least one word with a representation in a 1-hot-vector dictionary within the training window size. Figure 3 shows the model architecture.

## 3.2 Input Representation

In our character-level encoding, text is represented and processed at the granularity of individual characters. Traditional word- or token-level encodings break text into words or subwords and produce a one dimensional vector for each; by contrast, our character-level encoding yields a two-dimensional representation per word, with a one-hot vector for each character in the sequence. Sequences begin with a [BEG] token and end with an [END] token. The tokens are divided into the subgroups shown in table 1.

For the input representations we first split the words into characters and then apply modifiers. The order is always [UP] (if applicable), diacritic modifiers (if applicable), and then the base character.

| Category | Tokens |
|---|---|
| Base characters | a–z |
| Digits | 0–9 |
| Special symbols | ø, þ, ð, ł, ŋ, °, ˆ, !, ", §, $, %, &, /, (, ), =, ?, ‘, ´, +, *, ~, #, ’, _, ., :, ,, ;, <, >, @, €, {, [, ], }, . . . , „, ", ", ‚, ’, -, », « |
| Special word-level tokens | [BEG] (beginning of a word), [END] (end of a word), [UNK] (unknown character), [PAD] (padding token at the end of a word) |
| Special character-level tokens | Padding and unknown characters |
| Modifiers for diacritics | Acute Accent (´), Grave Accent (‘), Macron (¯), Tilde (~), Horn (˛), Diaeresis (¨), Ligature modifier, Uppercase modifier, Sharp S modifier (ß) with base character ’s’ |

Table 1: Token categories and their corresponding characters.

### 3.2.1 Method

Starting from raw text, we split the sequence into words, treating each word and special symbol as a separate token, resulting in our corpus $T$. Each word is converted character-by-character into its one-hot-vector representation using the predefined character-to-index mapping from our alphabet dictionary $A$. We then pad each word to maximum allowed length with a special character-level padding token. Special symbols and special tokens such as [CLS], [SEP], and [PAD] are treated as one-character words. Modifier tokens from $A$ are inserted as needed to encode diacritics, uppercase letters, and ligatures. Each word representation starts with [BEG] and ends with [END], followed by [PAD] tokens for batch processing if needed. When an uppercase character with a diacritic is encountered, we first add the [UP] modifier if needed, then a potential diacritic modifier, and finally the base character in lowercase. For example, *Liberté* is encoded as [BEG] [UP] [l] [i] [b] [e] [r] [t] [´] [e] [END] [PAD], with the number of [PAD] tokens depending on the maximum word length in the batch. The principle of splitting up characters into multiple tokens is also applied to ligatures and other special characters. The [END] token explicitly marks the end of the word for reconstruction tasks during training. In general, the special tokens are close to the usage of special tokens in BERT-like models (Devlin et al., 2019).

### 3.3 Model Architecture

RCE uses a classical transformer encoder model (Vaswani et al., 2017) to generate the word embeddings. The input is formed using the method described in Section 3.2, yielding a tensor of the size $|A| \times |C|$, where $|A|$ is the size of the alphabet dictionary and $|C|$ is the length of the character string plus the special tokens. A linear layer encodes this to size $\text{len}(e) \times |C|$. We then add positional encodings to each of the $|C|$ input embedding vectors. In contrast to typical transformer-encoder applications, we only use the first output (corresponding to [BEG]) as the word embedding, analogous to [CLS] in BERT-like models (Devlin et al., 2019).

### 3.4 Model Training

Analogous to traditional word embedding methods like word2vec and FastText, we train by predicting surrounding words for each word in the corpus, encouraging the resulting embeddings to encode semantic information. Additionally, we train the model to predict the character string of the input word itself, preserving syntactic information in the embedding.

### 3.4.1 Context Prediction

Similar to models like word2vec (Mikolov et al., 2013; Bojanowski et al., 2017), we learn a vector represention model by predicting the surrounding words. However, instead of predicting dictionary entries of the tokens in this configurations, we predict their character strings.

We take the word vector $e$ from our embedding model and replicate it $l$ times, with $l$ being the maximum word length. We then add positional embedding to those word vectors, in similar fashion to other transformer-based models. The resulting tensor is then fed to another transformer encoder model. The output is then passed through a linear layer to predict the one-hot encoded characters of the surrounding tokens, similar to the input. We choose an encoder rather than a decoder to allow end-to-end training without recursively predicting characters one by one. As with classic word2vec models, fully achieving the training target of perfectly predicting the surrounding tokens and their spelling is unnecessary, since the end goal of the training is an information-dense vector representation of the word, not actual context prediction. Since we can use this method to train the model also on rare, unusually spelled, or unusually inflected surrounding words, character-based context prediction enables our model to also learn from such cases. In short, the actual prediction accuracy is less important than the training signal.

$$L_{\text{CE}} = -\frac{1}{|C|} \sum_{p=1}^{|C|} \sum_{i=1}^{|A|} y_{p,i} \log \hat{y}_{p,i}. \tag{1}$$

For context prediction we use cross-entropy loss, defined in equation 1, to predict the characters of the surrounding words. Here, $|A|$ is the size of the alphabet dictionary, $|C|$ is the maximum character sequence length, $y_{p,i}$ is the ground-truth binary label for character class $i$ at position $p$, and $\hat{y}_{p,i}$ is the predicted probability for character class $i$ at position $p$. We apply a softmax activation so that the predicted character probabilities sum to 1, and average the loss over all characters in the predicted word.

For context words we use a window size of 5, meaning that for each word in the training data we predict a random word from among the five preceding and five following words.

### 3.4.2 Identity Prediction

Another training objective is to use the resulting word embedding vector $e$ to reconstruct the encoded input word itself. Under this objective, the model does not primarily learn the semantic meaning of the word but instead encodes the character string in a manner similar to an autoencoder. This yields a word vector that preserves syntactic information in addition to semantic information. The general training procedure is similar to the context prediction in Section 3.4.1. Identity prediction is intended to complement semantic training objectives, ensuring that syntactic information remains present in the resulting word vector.

For the identity prediction, we apply the same cross-entropy loss from equation 1 as for the context prediction in section 3.4.1, averaged over all characters in the predicted word.

### 3.4.3 Context Dictionary Prediction

Word2Vec (Mikolov et al., 2013) and FastText (Bojanowski et al., 2017) predict dictionary entries of context words or subwords while BERT predict masked dictionary tokens within the input sequence (Devlin et al., 2019). Although our approach does not rely on dictionary-based word encoding to generate word vector, the training signal from traditional dictionary entry prediction is a simpler task than the character-based context prediction explained in Section 3.4.1, making it an easier training task for the model. Therefore, predicting the surrounding tokens, if they appear in the dictionary, serves as our third training objective, in addition to the methods presented in Section 3.4.1 and 3.4.2.

For this objective, we use a cross-entropy loss similar to equation 1, but here $C$ is the size of the dictionary $|D_T|$ and $y_i$ and $\hat{y}_i$ denote the ground truth and predicted one-hot vector values of the dictionary entry of the chosen surrounding word. As with the other context prediction objective, we use a context window of 5, meaning that for each word in the training data we predict a random word from among the five preceding and five following words.

### 3.4.4 Combination of Training Objectives

The three training objectives presented in Section 3.4.1, 3.4.2, and 3.4.3 are combined in the training process.

$$L_{total} = L_{context} + L_{identity} + \frac{L_{dict}}{10} \tag{2}$$

Equation 2 defines the total loss $L_{total}$ as the sum of the context prediction loss $L_{context}$, the identity prediction loss $L_{identity}$, and the context dictionary prediction loss $L_{dict}$. We scale the dictionary prediction loss by a factor of $\frac{1}{10}$ because preliminary experiments showed it was otherwise much larger than the two other losses, leading to poorer results.

### 3.4.5 Hyperparameter Search

For the training objectives described in Section 3.4, we searched for suitable encoder hyperparameters. We used a limited training regime of 100000 training steps with a batch size of 512. To obtain robust general hyperparameters, we used a mixed corpus containing Icelandic, Norwegian, German, Latin, Faroese, Uzbek, and Middle High German. The hyperparameters we searched for were the learning rate $(0.1, 0.01, 0.001, 0.0001, 0.00001)$, the embedding size $(64, 128, 256)$, the size of the transformer feedforward layer $(64, 128, 256, 512)$, the number of encoder layers $(2, 3, 4)$, and the number of attention heads in the encoder $(2, 4, 8)$. We trained all combinations across the three training objectives (context prediction, identity prediction, and context dictionary prediction) from Section 3.4. We selected as hyperparameters a learning rate of 0.001, an embedding size of 64, 3 transformer encoder layers and 2 encoder attention heads.

## 4 Evaluation

Evaluating a word embedding model is different from evaluating models that are directly trained for tasks such as classification. Two distinct methodologies may be used: (1) analyzing the properties of the embedding vectors to ascertain their alignment with specified criteria, and (2) measuring the performance of the word embedding vectors on downstream tasks. Because analyzing the properties is particularly challenging for low-resource languages, we adopt the *odd-one-out (OOO)* and *TopK* methods (Stringham & Izbicki, 2020). For downstream evaluation we employ a variety of tasks, including the SWAG (Zellers et al., 2018) benchmark, declension prediction for inflected languages, and metaphor and chiasmus detection.

### 4.1 Experimental Setup

We pretrain our models on the fist 1,000,000 words of the CC100 corpus (Conneau et al., 2020) dataset for the respective language.

### 4.2 Baseline Models

As our baseline vector generation models, we use FastText and char2vec. As noted above, FastText extends the concept of word2vec by incorporating subword information. Because many modern static word embedding approaches also employ convolution-based character information, we use a simple implementation similar to char2vec (Ho, 2025), which we denote with c2v in our experiments. The character string representations we use for the char2vec model are the same as for the RCE model. In the c2v approach, a shared character projection is applied to the one-hot character representation at each character position, which can be represented as a position-wise $1 \times 1$ convolution over the character-feature dimension. The projected character sequence is then fed into a recurrent LSTM encoder, and the final hidden state is used as the word representation. Thus, both c2v and RCE derive word vectors from character strings, but they differ in encoder architecture: c2v uses a recurrent character encoder, whereas RCE uses a transformer encoder. Both are trained using explicit context, identity, and dictionary-prediction objectives. The comparison with c2v is therefore central to our evaluation, because it tests whether the proposed RCE architecture and objectives improve over a strong character-based baseline.

### 4.2.1 TopK and OddOneOut

We use *TopK* and *OddOneOut (OOO)* to analyze: (1) whether our embedding vectors exhibit the desired property that words appearing in similar contexts have similar embeddings, and (2) whether words from different contexts are represented distinctly. These two metrics are specifically designed for evaluating embeddings in low-resource languages (Stringham & Izbicki, 2020). Both require datasets organized into categories with their corresponding members. For example, the category *transportation* might contain *bus*, *car*, and *plane*, while *politics* could include *chancellor*, *president*, and *parliament*.

For the TopK metric, we compute vectors for all words in the evaluation dataset. For each word, we retrieve the top $k$ closest words based on Euclidean distance in the embedding space. We set $k = 3$ following Stringham & Izbicki (2020) in our experiments. The percentage of retrieved words belonging to the same category defines the score for that word. The dataset-level TopK score is the average across all words. A score of 1 indicates that every word is surrounded only by words of the same category, while a score of 0 means that all words are clustered with words from different categories.

For the OOO metric, we randomly select 10 words from one category and one word from a different category, as in Stringham & Izbicki (2020). We compute the mean vector of these 11 words and then identify the word farthest from this prototype. If the outlier is the word from the different category, the set receives a score of 1; otherwise, 0. The OOO score is the average across many such random sets—1000 in our case.

We constructed category-based evaluation datasets for Faroese, Latin, Middle High German, and Uzbek. The Faroese dataset contains 8 categories and 161 words; the Latin dataset contains 9 categories and 158 words; the Middle High German dataset contains 9 categories and 205 words; and the Uzbek dataset contains 9 categories and 141 words. The reported TopK and OOO experiments use the Faroese, Latin, and Uzbek datasets. These datasets are released together with our code.

### 4.2.2 Inflection

In contrast to English or Mandarin Chinese, many languages such as Latin are highly inflected. For example, the form of nouns changes depending on the case and number of the word (Ayer, 2014). In traditional methods this requires a separate dictionary entry for each form—unless tokenization methods such as WordPiece happen to separate the stem from the ending—leading to weaker representations of even the base form compared with approaches that exploit stem similarity, such as ours. Moreover, the information contained in inflectional endings should generalize to other words not seen during training. Just as a human familiar with a declension pattern can infer unseen forms from the base form (or vice versa), a model should also benefit from such regularities. The same principle applies to other types of inflection, such as conjugation.

To evaluate our method on inflection, we use a Latin declension prediction task: predicting the declension class of a noun from its nominative and genitive singular forms. The evaluation dataset contains 18,497 Latin noun entries and was generated from a Latin dictionary website[1]. We also retain a larger source/intermediate file during dataset construction, but this file is not used as the evaluation input.

### 4.2.3 Stylistic Device Detection

In stylometry, *stylistic device detection* has long been studied for some devices such as *metaphors*, while others such as the *chiasmus* have received only limited attention. Because both of these devices have also been analyzed in low-resource languages—where additional challenges arise—we compare our word vectors against those used in prior studies to highlight the performance of our approach.

**Chiasmus Classification**  For chiasmus classification we adopt the method of (Schneider et al., 2021). To determine whether a phrase constitutes a chiasmus, the authors employ various features, including pairwise distances of the embeddings of the four main words, yielding six vector-distance features. In our experiments we apply both our novel word embedding approach and the baseline methods. We evaluate on a German labeled chiasmus candidate dataset containing 4,447 entries and perform a 5-fold cross-validation.

---

[1] https://www.latin-is-simple.com

**Metaphor Classification** For metaphor classification we use the method for low-resource metaphor prediction proposed in (Schneider et al., 2022). Because this method is specifically designed for low-resource languages, we replace the word embeddings in their framework with our novel approach and with the baseline methods. We evaluate on a German metaphor-pair dataset containing 1,200 pairs with 1,200 corresponding labels.

For comparison, we trained both a BERT-like model with traditional WordPiece tokenization and a BERT-like model using our novel approach. In the latter, the [CLS], [SEP], [PAD], and [MASK] tokens of BERT are treated as ordinary words composed of their characters when used with our novel approach. We evaluated these models on a downstream experiment similar to SWAG (Zellers et al., 2018), using datasets constructed from the respective corpora. The datasets are released together with our code.

### 4.2.4 SWAG-like Sentence Continuation

For the BERT-like experiments, we construct a SWAG-like multiple-choice sentence continuation dataset from CC100-style text for English, German, Faroese, Norwegian, Uzbek, and Icelandic. Each example contains a context sentence, a partial continuation, four candidate endings, and a label indicating the correct ending. Sentences are split on punctuation marks (., !, ?, ;, and :). We discard sentences that start with a lowercase or non-alphanumeric character, contain fewer than 5 words, have fewer than 10 characters, or have more than 120 characters.

Each example is generated from two consecutive valid sentences. The first sentence is used as the context. The first two words of the second sentence form the partial continuation, and the remaining words of that sentence form the correct ending. The three distractor endings are sampled from other examples. The label indicates the position of the correct ending among the four candidate endings.

The resulting train and validation splits contain 330,000 examples in total, with 264,000 training examples and 66,000 validation examples. Table 2 reports the per-language dataset sizes.

| Language | Train | Validation | Total |
|----------|-------|------------|-------|
| English | 80,000 | 20,000 | 100,000 |
| German | 80,000 | 20,000 | 100,000 |
| Norwegian | 80,000 | 20,000 | 100,000 |
| Faroese | 8,000 | 2,000 | 10,000 |
| Icelandic | 8,000 | 2,000 | 10,000 |
| Uzbek | 8,000 | 2,000 | 10,000 |
| Total | 264,000 | 66,000 | 330,000 |

Table 2: Dataset sizes for the custom SWAG-like sentence continuation task.

As hyperparameters we selected a feedforward layer size (general tensor size) of 128, four attention heads, and four encoder layers for the language transformer. For the character-based version we used an embedding size of 128 and two layers each for the character-token encoder and decoder. The batch size was 12 sentence pairs per batch. Each pair consisted of two sentences from the training corpus, with the second sentence being either random or the true continuation of the first, accompanied by a label indicating whether the continuation is correct. We trained both model types with next-sentence prediction and masked language modeling, replacing 15% of the input tokens with [MASK]. For optimization we used the Adam optimizer with a maximum learning rate of 0.001, scheduled by a linear warmup over the first 5,000 steps followed by cosine annealing, for a total of 300,000 steps. We set the dropout rate was to 0.1.

### 4.3 Results

### 4.3.1 TopK and OddOneOut

We manually created a Faroese dataset with 20 categories and an average of 20 members per category. Models were trained on Faroese, Icelandic, and Norwegian corpora and evaluated on the Faroese set. TopK and

| target language | training language | RCE | c2v | RCE+c2v | fastText |
|---|---|---|---|---|---|
| Faroese | Faroese | 0.26 | 0.20 | 0.25 | 0.21 |
| Faroese | Norwegian | 0.22 | 0.20 | 0.24 | 0.24 |
| Faroese | Icelandic | 0.24 | 0.22 | 0.26 | 0.22 |
| Latin | Latin | 0.23 | 0.25 | 0.27 | 0.25 |
| Latin | Italian | 0.16 | 0.26 | 0.24 | 0.15 |
| Uzbek | Uzbek | 0.20 | 0.25 | 0.25 | 0.20 |

Table 3: This table shows the TopK results across target languages (Faroese, Latin, Uzbek) for models trained on the listed training languages. We compare RCE, c2v, RCE+c2v, and fastText. Because the entire test sets were used without splits, standard deviations cannot be reported.

| target language | training language | RCE | c2v | RCE+c2v | fastText |
|---|---|---|---|---|---|
| Faroese | Faroese | 0.13 | 0.12 | 0.29 | 0.08 |
| Faroese | Norwegian | 0.22 | 0.29 | 0.24 | 0.09 |
| Faroese | Icelandic | 0.11 | 0.14 | 0.16 | 0.09 |
| Latin | Latin | 0.25 | 0.60 | 0.30 | 0.10 |
| Latin | Italian | 0.18 | 0.35 | 0.31 | 0.08 |
| Uzbek | Uzbek | 0.17 | 0.36 | 0.39 | 0.10 |

Table 4: This table shows the OddOneOut (OOO) results across target languages (Faroese, Latin, Uzbek) for models trained on the listed training languages. We compare RCE, c2v, RCE+c2v, and fastText. Because the entire test sets were used without splits, standard deviations cannot be reported.

OOO results are reported in Table 3 and Table 4. Overall, character-based models (RCE and c2v) match or surpass fastText on both metrics, with the strongest scores from the combined RCE+c2v model. For TopK, RCE (alone and combined) shows clear gains; notably, when trained on Icelandic, RCE-based models outperform fastText trained directly on Faroese. For OOO, the character-based approaches are stronger still: the combined RCE+c2v model outperforms the others by a large margin, while c2v slightly exceeds RCE when used alone. These results indicate that our method yields embeddings more aligned with desired semantic structure than traditional token-based approaches such as fastText, particularly in low-resource settings.

Because RCE+c2v often performs as well as or better than models trained directly on the target language—even when trained on a related language—we infer that our approach captures semantic information more effectively than fastText.

### 4.3.2 Declension Detection by Inflection

| training language | RCE | c2v | RCE+c2v | fastText |
|---|---|---|---|---|
| Latin | $0.85 \pm 0.02$ | $0.90 \pm 0.01$ | $0.90 \pm 0.01$ | $0.85 \pm 0.02$ |
| Italian | $0.85 \pm 0.01$ | $0.88 \pm 0.02$ | $0.84 \pm 0.01$ | $0.50 \pm 0.01$ |

Table 5: This table shows the results for the declension prediction on a Latin dataset, comparing RCE, c2v, RCE+c2v, and fastText.

To evaluate performance in inflected languages, we conducted two experiments on Latin and Italian. We constructed a Latin dataset from a dictionary, categorizing 18,497 nouns by declension. Because a noun's declension can be inferred from its nominative and genitive singular forms (Ayer, 2014), we used those two

vectors as input to a simple feedforward neural network with one hidden layer, predicting declension from their concatenation. We compared RCE with c2v and fastText (Table 5).

Both c2v (with our training setup) and RCE perform well when pretrained on Latin or Italian, consistently outperforming fastText; c2v slightly outperforms RCE on this task. Notably, fastText is stronger when pretrained on the target language (Latin) but degrades substantially when pretrained on a related language (Italian). This suggests that character-level approaches capture inflectional information more robustly than traditional token-based methods such as fastText.

### 4.3.3 Stylistic Device Detection

We evaluate metaphor detection and chiasmus detection.

**The Chiasmus** is a stylistic device in which a phrase is followed by its inverted counterpart. Chiasmi are often used to emphasize contrasts between the two parts. For example: **long** *is the* **research***, but the* **paper** *is* **short**. (Schneider et al., 2021) proposed a detection method based on multiple features, including pairwise cosine distances between the embeddings of the four main words (here: *long*, *research*, *paper*, *short*).

|  | RCE | c2v | RCE+c2v | fastText |
|---|---|---|---|---|
| all features | $0.34 \pm 0.17$ | $0.32 \pm 0.13$ | $0.30 \pm 0.11$ | $0.31 \pm 0.13$ |
| only embedding | $0.19 \pm 0.06$ | $0.02 \pm 0.00$ | $0.32 \pm 0.14$ | $0.02 \pm 0.00$ |

Table 6: Results for the chiasmus detection experiments. We compare the results of RCE, c2v, and FastText.

We trained RCE, c2v, and fastText on a German corpus and evaluated on a German dataset using 5-fold cross-validation. Table 6 reports the results. With the full feature set of (Schneider et al., 2021), all models perform similarly. When restricting to embedding-only features, the combined RCE+c2v model performs best, followed by RCE; both c2v alone and fastText perform poorly. This suggests that recurrent (c2v) and transformer-based (RCE) encoders capture complementary information that is beneficial when combined.

**The Metaphor** is a stylistic device that uses a word or phrase from one domain in another to create a new meaning (e.g., *the sun is smiling*). (Schneider et al., 2022) proposed detecting adjective–noun metaphors (e.g., *moody weather*, *thirsty car*) by mapping adjective and noun embeddings into a metaphoricity space in which their cosine distance reflects metaphoricity. Both vectors are transformed by the same model.

| language | RCE | c2v | RCE+c2v | fastText |
|---|---|---|---|---|
| German | $0.73 \pm 0.05$ | $0.71 \pm 0.05$ | $0.74 \pm 0.04$ | $0.67 \pm 0.04$ |

Table 7: This table shows the results for the metaphor detection experiments. We compare the results of RCE, c2v, RCE+c2v, and FastText.

Table 7 reports metaphor detection results. We trained RCE, c2v, RCE+c2v, and fastText on a German corpus and evaluated on a German dataset using 10-fold cross-validation. Character-based models outperform fastText; RCE outperforms c2v; and the combined RCE+c2v model achieves the highest score.

### 4.3.4 Ablation of Training Objectives

We further ablate the training objectives used for the character-based models: context reconstruction (*ctx*), identity reconstruction (*id*), and dictionary prediction (*dict*). Detailed results are reported in Tables 9, 10, 11, and 12.

The ablations show that the usefulness of each training signal depends on the downstream task. For declension prediction, identity reconstruction is consistently important, which is expected because the task depends strongly on word-internal form and suffix information. In contrast, context reconstruction or dictionary prediction alone yields substantially weaker balanced accuracy in several settings.

For chiasmus detection, most objective combinations lead to majority-class behavior, while combinations including identity and dictionary prediction produce non-trivial MAP scores. For metaphor detection, dictionary-based objectives and combinations with identity reconstruction perform best, whereas context reconstruction alone is weak. The TopK and OddOneOut ablations show a less uniform pattern across languages, although dictionary-based and combined objectives are generally strongest for Faroese and Uzbek. Overall, these results indicate that the auxiliary training objectives contribute differently across tasks, and that combining form-preserving and lexical prediction signals is often beneficial.

### 4.3.5 BERT-like Model

| Model | Finetuning | Target | BERT-like | RCE-BERT-like (ours) |
|-------|-----------|--------|-----------|----------------------|
| en | en | en | 0.561 | **0.578** |
| de | de | de | 0.542 | **0.628** |
| no | no | no | 0.537 | **0.561** |
| is | is | is | 0.353 | **0.391** |
| fo | fo | fo | *0.279* | 0.265 |
| no | no | fo | 0.277 | **0.285** |
| no | fo | fo | *0.266* | 0.256 |
| is | is | fo | 0.244 | *0.280* |
| is | fo | fo | *0.275* | 0.241 |

Table 8: Results of the BERT-like model and the RCE-BERT-like model on the SWAG-like task. On the left side the language the base model was trained on is denoted as *Model*, the language of the task it was fine-tuned on is denoted as *Finetuning*, and the target language of the experiment is denoted as *Target*. The results are the accuracy on the test set; RCE-BERT-like is our approach. The upper four lines show the experiments on various larger languages. The lower five values show tests on the low-resource Faroese language, with various pretrainings.

Table 8 presents SWAG-like results for WordPiece-based vs. RCE-based BERT-like models. First, for English, German, Norwegian, and Icelandic, we pretrained and fine-tuned, and evaluated within the same language; in all four cases, the RCE-based model outperformed the WordPiece baseline.

Second, we evaluated transfer to the low-resource Faroese language. We pretrained on Faroese, Norwegian, or Icelandic; fine-tuned on either Faroese or the pretraining language; and evaluated on Faroese. The best Faroese result was achieved by the RCE-based model pretrained and fine-tuned on Norwegian, followed by the configuration pretrained and fine-tuned on Icelandic. These findings suggest that RCE-based embeddings can facilitate cross-language transfer in BERT-like models for low-resource settings.

## 5 Limitations

Our approach computes word embeddings directly from character strings and shows promising results for alphabet-based writing systems. However, it is developed for linear, one-dimensional scripts (e.g., Latin). This limits applicability to non-alphabetic or more complex systems—such as Mandarin Chinese or Japanese—where characters comprise multiple subcomponents arranged in two dimensions (Wu et al., 2025). While the underlying principles may be adaptable, additional modeling is required to capture such structure.

The present implementation relies on a predefined set of Latin characters and associated diacritics. Thus, it does not directly support other alphabet-based systems such as Cyrillic, Devanagari, or Hangul. Extending to these scripts would require redesigning the character–token mapping and the encoding step.

Finally, our evaluations are constrained by the range of languages and datasets considered and by available training hardware. Future work should scale evaluations across more languages and leverage additional compute to assess generalization more comprehensively.

## 6 Conclusion and Outlook

In this work we proposed **Rich Character Embedding** (RCE), a method for deriving word embeddings directly from character strings for low-resource and morphologically rich languages. RCE preserves a single vector per word while incorporating orthographic and morphological information that token- and subtoken-based approaches may fail to capture consistently.

Across multiple downstream tasks, RCE and the combined RCE+c2v variant matched or outperformed traditional baselines such as fastText, and RCE-based embeddings improved the performance of BERT-like models in several settings. The comparison with c2v shows that recurrent character encoders are a strong baseline; however, the complementary behavior of c2v and RCE suggests that transformer-based character encoders capture useful information that is not fully covered by recurrent character models alone.

The ablation study further shows that the training objectives contribute differently across tasks. Identity reconstruction is particularly useful for inflection-sensitive tasks such as declension prediction, while dictionary-based and combined objectives are important for metaphor and chiasmus detection. These results suggest that character-based embeddings benefit from combining form-preserving and lexical prediction signals rather than relying on a single objective.

Future work should extend the evaluation to more languages, larger corpora, and additional downstream tasks. Another important direction is adapting the character representation beyond Latin-script alphabets, where writing systems may require different assumptions about character decomposition and visual structure. Finally, the interaction between RCE and other character-aware or definition-based methods remains promising, especially for rare words and low-resource languages where both orthographic regularities and lexical resources may provide complementary information.

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

# A   Appendix

| Language | Model | Training Heads | TopK | OddOneOut |
|---|---|---|---|---|
| fo | RCE | ctx | 0.1300 | 0.0800 |
| fo | RCE | id | 0.2300 | 0.1800 |
| fo | RCE | dict | 0.1900 | 0.1400 |
| fo | RCE | id+ctx | 0.2400 | 0.1500 |
| fo | RCE | id+dict | 0.2600 | 0.1100 |
| fo | RCE | ctx+dict | 0.1500 | 0.1000 |
| fo | c2v | ctx | 0.1600 | 0.0700 |
| fo | c2v | id | 0.2700 | 0.1700 |
| fo | c2v | dict | 0.3200 | 0.3000 |
| fo | c2v | ctx+dict | 0.3300 | 0.2900 |
| fo | mixed | ctx | 0.1200 | 0.1200 |
| fo | mixed | id | 0.2700 | 0.2000 |
| fo | mixed | dict | 0.3400 | 0.2800 |
| fo | mixed | id+ctx | 0.2800 | 0.2100 |
| fo | mixed | id+dict | 0.2900 | 0.2400 |
| fo | mixed | ctx+dict | 0.3200 | 0.3300 |
| la | RCE | ctx | 0.1200 | 0.1100 |
| la | RCE | id | 0.2400 | 0.3700 |
| la | RCE | dict | 0.1300 | 0.1300 |
| la | RCE | id+ctx | 0.2700 | 0.4700 |
| la | RCE | id+dict | 0.2400 | 0.2100 |
| la | RCE | ctx+dict | 0.1300 | 0.1100 |
| la | c2v | ctx | 0.2100 | 0.4200 |
| la | c2v | id | 0.2900 | 0.4200 |
| la | c2v | dict | 0.2400 | 0.3900 |
| la | c2v | id+dict | 0.2100 | 0.3500 |
| la | c2v | ctx+dict | 0.2400 | 0.3500 |
| la | mixed | ctx | 0.2300 | 0.5700 |
| la | mixed | id | 0.2600 | 0.3400 |
| la | mixed | dict | 0.2500 | 0.4000 |
| la | mixed | id+ctx | 0.2700 | 0.3600 |
| la | mixed | id+dict | 0.2100 | 0.2200 |
| la | mixed | ctx+dict | 0.2500 | 0.3700 |
| uz | RCE | ctx | 0.0900 | 0.0900 |
| uz | RCE | id | 0.2600 | 0.3200 |
| uz | RCE | dict | 0.1500 | 0.2000 |
| uz | RCE | id+ctx | 0.2000 | 0.2300 |
| uz | RCE | id+dict | 0.2500 | 0.2500 |
| uz | RCE | ctx+dict | 0.1400 | 0.2100 |
| uz | c2v | ctx | 0.1300 | 0.3000 |
| uz | c2v | id | 0.2500 | 0.3800 |
| uz | c2v | dict | 0.3500 | 0.4900 |
| uz | c2v | id+ctx | 0.2200 | 0.2200 |
| uz | c2v | id+dict | 0.2600 | 0.3600 |
| uz | c2v | ctx+dict | 0.3400 | 0.4100 |
| uz | mixed | ctx | 0.3000 | 0.4800 |
| uz | mixed | id | 0.2500 | 0.2000 |
| uz | mixed | dict | 0.3600 | 0.5100 |
| uz | mixed | id+ctx | 0.2600 | 0.2600 |
| uz | mixed | id+dict | 0.2700 | 0.3600 |
| uz | mixed | ctx+dict | 0.3700 | 0.4400 |

Table 9: Ablation results for the TopK and OddOneOut evaluations on Faroese, Latin, and Uzbek. *ctx* stands for context reconstruction, *id* stands for identity reconstruction, *dict* stands for one-hot dictionary word prediction. The best objective combination varies across languages and metrics: dictionary-based and combined objectives are strongest for Faroese and Uzbek, while Latin shows more varied results for TopK and OddOneOut.

| Training Language | Model | Training Head | Balanced test acc. |
|---|---|---|---|
| it | RCE | ctx | $0.1622 \pm 0.0016$ |
| it | RCE | id | $0.8658 \pm 0.0178$ |
| it | RCE | dict | $0.2097 \pm 0.0040$ |
| it | RCE | id+ctx | $0.8537 \pm 0.0236$ |
| it | RCE | id+dict | $0.8821 \pm 0.0186$ |
| it | RCE | ctx+dict | $0.1483 \pm 0.0020$ |
| it | c2v | ctx | $0.5741 \pm 0.0109$ |
| it | c2v | id | $0.9045 \pm 0.0096$ |
| it | c2v | id+ctx | $0.8833 \pm 0.0091$ |
| it | c2v | id+dict | $0.9000 \pm 0.0088$ |
| it | c2v | ctx+dict | $0.7258 \pm 0.0100$ |
| it | mixed | ctx | $0.1845 \pm 0.0019$ |
| it | mixed | id | $0.9012 \pm 0.0119$ |
| it | mixed | dict | $0.7419 \pm 0.0219$ |
| it | mixed | id+dict | $0.8909 \pm 0.0224$ |
| it | mixed | ctx+dict | $0.7442 \pm 0.0253$ |
| la | RCE | ctx | $0.1430 \pm 0.0019$ |
| la | RCE | id | $0.8858 \pm 0.0122$ |
| la | RCE | dict | $0.1506 \pm 0.0020$ |
| la | RCE | id+ctx | $0.8711 \pm 0.0275$ |
| la | RCE | id+dict | $0.8644 \pm 0.0122$ |
| la | RCE | ctx+dict | $0.1971 \pm 0.0013$ |
| la | c2v | ctx | $0.4182 \pm 0.0011$ |
| la | c2v | id | $0.8895 \pm 0.0054$ |
| la | c2v | dict | $0.3993 \pm 0.0065$ |
| la | c2v | id+dict | $0.8821 \pm 0.0176$ |
| la | c2v | ctx+dict | $0.5132 \pm 0.0096$ |
| la | mixed | ctx | $0.2819 \pm 0.0012$ |
| la | mixed | id | $0.9064 \pm 0.0202$ |
| la | mixed | dict | $0.4918 \pm 0.0061$ |
| la | mixed | id+ctx | $0.9000 \pm 0.0156$ |
| la | mixed | id+dict | $0.9052 \pm 0.0209$ |
| la | mixed | ctx+dict | $0.4705 \pm 0.0031$ |

Table 10: Ablation study for declension prediction. *ctx* stands for context reconstruction, *id* stands for identity reconstruction, *dict* stands for one-hot dictionary word prediction. Across model types, context prediction and dictionary prediction alone provide a weak training signal, while identity reconstruction provides a strong training signal. This is expected since the information about the declension can be mostly inferred from the word suffix, which is explicitly reconstructed in the identity prediction head. This is also visible in the combinations: context and dictionary without identity provide the weakest training signal, similar to training with context or dictionary prediction alone.

| Model | Training Heads | MAP |
|-------|----------------|-----|
| RCE | ctx | $0.0180 \pm 0.0000$ |
| RCE | id | $0.0180 \pm 0.0000$ |
| RCE | dict | $0.0180 \pm 0.0000$ |
| RCE | id+ctx | $0.0180 \pm 0.0000$ |
| RCE | id+dict | $0.1852 \pm 0.0811$ |
| RCE | ctx+dict | $0.0180 \pm 0.0000$ |
| c2v | ctx | $0.0180 \pm 0.0000$ |
| c2v | dict | $0.0180 \pm 0.0000$ |
| c2v | id+ctx | $0.0180 \pm 0.0000$ |
| c2v | id+dict | $0.2645 \pm 0.1141$ |
| c2v | ctx+dict | $0.0180 \pm 0.0000$ |
| mixed | ctx | $0.0180 \pm 0.0000$ |
| mixed | id | $0.0180 \pm 0.0000$ |
| mixed | dict | $0.0180 \pm 0.0000$ |
| mixed | id+ctx | $0.0180 \pm 0.0000$ |
| mixed | id+dict | $0.2153 \pm 0.0500$ |

Table 11: Results for the chiasmus detection ablation study. *ctx* stands for context reconstruction, *id* stands for identity reconstruction, *dict* stands for one-hot dictionary word prediction. In this experiment, most training combinations that do not use all training heads yield only majority-class prediction behaviour, suggesting that the complex task of chiasmus detection requires a combination of training signals. However, the combination of identity prediction and dictionary word prediction provided a rich enough training signal in this case.

| Model | Training Heads | Test AP |
|-------|----------------|---------|
| RCE | ctx | $0.4535 \pm 0.0223$ |
| RCE | id | $0.6178 \pm 0.0508$ |
| RCE | dict | $0.4836 \pm 0.0476$ |
| RCE | id+ctx | $0.6138 \pm 0.0576$ |
| RCE | id+dict | $0.6920 \pm 0.0660$ |
| RCE | ctx+dict | $0.4935 \pm 0.0480$ |
| c2v | ctx | $0.4375 \pm 0.0005$ |
| c2v | dict | $0.7275 \pm 0.0453$ |
| c2v | id+ctx | $0.6784 \pm 0.0644$ |
| c2v | id+dict | $0.7274 \pm 0.0746$ |
| c2v | ctx+dict | $0.7074 \pm 0.0526$ |
| mixed | ctx | $0.4252 \pm 0.0328$ |
| mixed | id | $0.6399 \pm 0.0733$ |
| mixed | dict | $0.7060 \pm 0.0638$ |
| mixed | id+ctx | $0.6677 \pm 0.0469$ |
| mixed | id+dict | $0.7258 \pm 0.0336$ |

Table 12: Ablation results for German metaphor detection, reported as test average precision. *ctx* stands for context reconstruction, *id* stands for identity reconstruction, *dict* stands for one-hot dictionary word prediction. Context reconstruction alone yields weak performance, while dictionary-based objectives and combinations with identity reconstruction provide the strongest results across model variants.

