# OpenReview forum: "Beyond Subtokens: A Rich Character Embedding for Low-Resource and Morphologically Complex Languages"
_TMLR — Accepted by TMLR_

### Review · Reviewer_e5UR · 2026-04-02

**Summary Of Contributions:**

This paper explores replacements for word-embedding models (mostly non-contextual variants like word2vec, Glove, FastText, but with aspiration for BERT, GPT models) using embeddings that are more based on characters.  The paper has a lot of care in how they designed the embeddings, especially to handle inflection.  The target use case is low-resource languages where there may not be enough text to build reliable embeddings otherwise, but there may be enough from this character-level approach.

The paper compares on several scenarios of low-resource language against FastText and some char2vec software (on GitHub for 4 or maybe 7 years).  The char2vec seems to do the best, but sometimes concatenating char2vec and their method provides some improvement.

**Additional Comments:**

There are a number of consistent misspellings.  In particular
  -  "trunsformer" -> "transformer"
  - "rundom" -> "random"

each appear several times.  Please check all of these, and then run a spell check.


The paper also felt repetitive at the end.  I think Sections 4.2.2, 4.2.3, 4.2.4 in the experimental set up could have been combined with the results on this topics.

**Audience:**

Yes

**Audience Explanation:**

Alternatives to language embeddings are interesting, and are worth exploring since they are of such high important in modern ML.
And tasks for low resource languages an area within this topic where it is hard to make progress so even limited progress with fresh new ideas is useful.

**Broader Impact Concerns:**

No concerns.  The paper is addressing important broader impact work in low resource languages.

**Claims And Evidence:**

No

**Claims Explanation:**

I found most of the paper well done and interesting to read.  It did a nice job of explaining the method and motivating it.  The choice of evaluations is solid experimentally, and of high interest because it targets several diverse (latin-based) low resource languages.

The problem is the char2vec method (which has been around for a number of years -- although I do not know any papers, other than the earlier cited one by Cao (AAAI 16) which compares to it), is introduced very late, and not explain well.  Char2Vec is apparently very similar to the approaches in this paper RCE, and performs slightly better.

There are some experiments where concatenating char2vec with RCE that outperforms either, but the results are not universal -- sometimes this leads to large decrease in performance compared to char2vec.

For this paper to be of use for the community, I believe it should:
  - explain more clearly what char2vec does
  - how their method RCE differs structurally to char2vec
  - identify structural advantages RCE has over char2vec


I would like to see this paper get in shape to publish.  Even if it was a paper that explained well when char2vec models and variants worked well in these context (including importantly, giving a clear description of char2vec in paper form, plus their however minor improvements) I could potentially support it.

**Requested Changes:**

See the discussion in the **Claims** section.  Please clearly explain:
  - how char2vec works
  - how your approach differs from it
  - perhaps an ablation study of the design choices in char2vec and RCE

If the authors complete the above, I think this will make a very useful contribution.

---

> ### Author Response · Authors · 2026-04-30
> **Answer to reviewer e5UR**
>
> Thank you very much.
> In the revised version we will explain c2v earlier in the paper and introduce it more properly, and highlight the differences between c2v and RCE.
> We are currently training the RCE and c2v models for the ablation study with different prediction heads. Unfortunately the training is not finished in time for this review, but the full ablation study will be in the final paper.
>
> For TopK and OOO, we have the results for RCE with only identity and dictionary prediction head and for c2v only the context prediction head.
> * Best TopK in paper: 0.26, RCE identity and dictiontary: 0.23, c2v with context:  0.16
> * Best OOO in paper: 0.29, RCE identity and dictionary: 0.12, c2v with context: 0.07
>
> For the latin declension prediction we have for Italian: RCE with identity and context, RCE with dictionary, c2v with context and dictionary. For Latin we have c2v with identity and dictionary and mixed with identity.
> * Best paper results: Latin: 0.90+-0.01, Italian: 0.88+-0.02
> * Italian RCE identity context: 0.84+-0.03
> * Italian RCE dictionary: 0.21+-0.00
> * Italian c2v context dictionary: 0.73+-0.01
> * Latin c2v identity dictionary: 0.88+-0.01
> * Latin mixed identity: 0.90+-0.01
>
> There are some interesting results here already; the dictionary prediction seems not to be enough for declension prediction, but  identity prediction with the mixed model seems to work well.
>
> For chiasmus and metaphor detection in German we have:
> Chiasmus:
> *  best in paper: 0.32+-0.14
> *  RCE with identity:  0.22+-0.09
> * c2v with identity and dictionary: 0.26+-0.11
> * c2v with dictionary: 0.02+-0.00
>
> Metaphor:
> * best in paper: 0.74+-0.04
> * RCE with identity: 0.69+-0.05
> * c2v with identity and dictionary: 0.74+-0.04
> * c2v with dictionary: 0.72+-0.07
>
> In the next version of the paper we will include the full ablation study with a discussion of the results and implications.
> We will also improve the spelling and see how we can restructure the experimental setup.

---

### Review · Reviewer_TXWK · 2026-04-09

**Summary Of Contributions:**

The paper proposes Rich Character Embeddings, a method for generating word vectors directly from character strings rather than from token or subtoken dictionaries. The approach uses a Transformer-based architecture to map a word's character sequence into a single dense vector. RCE is designed to simultaneously capture semantic information via context prediction and syntactic information via word reconstruction. The model aims to address the limitations of classical subtokenization in low-resource and morphologically complex languages, serving as a drop-in replacement for existing embedding layers.


**Strengths**

- Morphology: captures orthographic similarities, proving particularly robust for highly inflected languages ​​where subtoken models often fail to connect common roots.

- Efficiency: Unlike purely character- or byte-based models that drastically lengthen the input sequence, RCE maintains a single vector per word, preserving the context space for the main model.

- Applicability: Improves both small models (word2vec) and large context architectures (BERT), outperforming baselines like FastText in low-resource tasks.

**Weaknesses**

- The current implementation is restricted to Latin-based alphabets, excluding non-alphabetic writing systems (Chinese, Japanese). This is not a serious limitation, but it would be helpful to justify the choice to avoid misunderstandings.

- Consequence of the first point: Experimental evaluations are limited to a limited number of languages ​​and datasets, leaving uncertainty about generalisation across scales.

- Structurally, however, Dependence on Modifiers: The system of encoding diacritics and capitalisation via modifiers adds complexity to the input phase and may not be optimal for all orthographic varieties. It would be very useful to discuss this point.


To summarise, the paper is well written, and the methodology is clear, though there are some issues that do not limit its quality.

**Audience:**

Yes

**Audience Explanation:**

I believe the paper is of interest to TMLR's audience.
Points to be appreciated include Overcoming sub-tokenisation, which offers a valid alternative to the limitations of models like WordPiece and BERT in morphologically rich languages.
Support for low-resource languages demonstrates how to generate effective embeddings even with limited corpora, outperforming baselines such as FastText.
Beyond the merits, there is Architectural efficiency: it introduces a module that captures orthographic similarities without the sequence-length explosion typical of character models.

**Broader Impact Concerns:**

No critical ethical concerns emerge; however, the paper would benefit from a brief discussion of the potential positive social impact of bridging the digital divide for underrepresented linguistic communities. However, the authors should mention the risk of technological bias arising from the current limitation to the Latin alphabet, which could exclude several low-resource languages ​​that use other writing systems. Finally, the model's ability to operate on small corpora suggests a more sustainable environmental footprint than traditional large-scale models.

**Claims And Evidence:**

Yes

**Claims Explanation:**

First, the model is successfully tested on morphologically complex and low-resource languages ​​such as Latin.
The results show that RCE consistently outperforms FastText and WordPiece in specific metrics for limited data (TopK, Odd-One-Out).
The paper demonstrates that RCE captures superior syntactic information, as explained by experiments on Latin declension prediction.
Finally, clear evidence is provided of RCE's validity as a replacement in BERT-like architectures, improving its performance in cross-lingual contexts.

**Requested Changes:**

Authors should include a section explicitly discussing the model's weaknesses, particularly its limitations to Latin alphabets and the need for larger-scale testing. Clarifications requested during the review should be included in the final text to ensure full transparency regarding the evaluation criteria and architecture.

**Adjustments**

Finally, it would be helpful to include a Comparison with the State of the Art. We suggest citing the work by Ruzzetti et al. (2022), "Lacking the Embedding of a Word? Look It Up Into a Traditional Dictionary," to enrich the discussion of rare words (OOV). While RCE solves the problem through character structure, Ruzzetti et al. use external definitions; a comparison between an "orthographic" and a "definitional" approach would further deepen the paper's positioning within the landscape of solutions for low-resource languages.

---

> ### Author Response · Authors · 2026-04-30
> **Answer to reviewer TXWK**
>
> Thank you for this review.
> We will include a comparison with Ruzzetti et al. for the languages where we have dictionaries available in the next version of our paper. We will also expand upon describing the weaknesses and the limited experiments/the need for larger-scale testing.
> Additionally we will include a discussion of the potential positive social impact and about the limitation of the impact because of the current restriction on Latin script.

---

### Review · Reviewer_Mb7X · 2026-04-16

**Summary Of Contributions:**

This paper presents a method for learning word embeddings (for Latin-alpahet, whitespace-based languages) from character embeddings.  This should overcome certain limitations of word and subword embeddings (imperviousness to spelling-relatedness, morphology, etc) and at the same time overcome limitations of doing full byte or character-level modeling (longer sequences, harder learning problem).  The model first learns a transformer encoder on character strings, trained to predict both adjacent words' character strings as well as word identities (a la word2vec).  Through a series of evaluations, they show that the resulting word embeddings have some advantage over methods like word2vec and fasttext, and that it improves performance in a BERT-like model when used instead of subword tokenization (though see below for some hesitations with this experiment).

**Audience:**

Yes

**Audience Explanation:**

Improving tokenization for low-resource languages is an important problem in NLP that will be of interest to some TMLR readers.

**Claims And Evidence:**

Yes

**Claims Explanation:**

While the evidence appears good, I would put some marks against "clear": more detail needs to be provided about dataset construction and information, since the paper makes heavy use of custom datasets, especially for evaluation.

**Requested Changes:**

- The evaluation datasets are extremely under-specified.  Please provide more detailed information about created datasets (e.g. the Latin declension one and your custom SWAG-like dataset).
- I would like a comparison---both descriptively and empirically---with a hierarchical Transformer approach.  This paper compares using RCE embeddings in a BERT model with using subword tokens.  In the latter case, however, the subword embeddings are learned during the BERT pre-training.  Why. not compare with also learning the [BEG] RCE embeddings during the BERT pretraining, instead of separately first?  This would also make your model look more like a hierarchical transformer, which is why I'd want a discussion of that too.

Typographic items:
- Throughout the paper, "trunsform" is written instead of "transform" in several places and variations of the word (e.g. "trunsformer", "trunsformed").  Similarly, "appearunce" instead of "appearance", "runge" instead of "range", and "rundom" instead of "random".
- Hyphens are used in many places where em-dashes (`---` in TeX) should be used.

---

> ### Author Response · Authors · 2026-04-30
> **Answer to reviewer Mb7X**
>
> Thank you for this review.
> From the requested changes, I see that we may have not explained our approach clear enough. for the RCE embeddings in the BERT-like transformer, we already train the embeddings and the larger transformer jointly, no pre-trained RCE is used. However, we can can also add additional experiments with pre-trained RCE plugged into the BERT-like model.
> About the datasets: We will revise this and add more details in the next version of the paper.
> We will also fix the spelling and em-dashes.

---

### Decision · Action_Editor_UpH4 · 2026-05-25

**Recommendation:** Accept with minor revision

**Additional Comments:**

While the reviewers found the paper interesting, the authors failed to provide a revised version including all the promised changes in the responses.  Thus, a revised version including all the promised changes must be provided and not just submitting the current version.  In particular, reviewer e5UR recommended the acceptance on the condition of seeing a proper explanation of how Char2Vec works and compares to their RCE.

**Audience:**

Yes

**Audience Explanation:**

Improving tokenization for low-resource languages is an important problem in NLP that will be of interest to some TMLR readers.

**Claims And Evidence:**

Yes

**Claims Explanation:**

The reviewers find the paper's proposed RCE method promising and well-evaluated for morphologically complex, low-resource languages, noting that it successfully captures syntactic information and outperforms baselines like FastText and WordPiece. However, they emphasize that for the paper to be published, the authors must address a major flaw: the failure to adequately explain and structurally compare RCE against a similar, sometimes better-performing existing method char2vec.